# Extrafine HFA-beclomethasone-formoterol vs. nonextrafine combination of an inhaled corticosteroid and a long acting β2-agonist in patients with persistent asthma: A systematic review and meta-analysis

**Ting Liu[☉], Dan Yang[☉], Chuntao Liu[iD]***

Department of Respiratory and Critical Care Medicine, West China School of Medicine and West China Hospital, Sichuan University, Chengdu Province, China

[☉] These authors contributed equally to this work.

* taosen666999@163.com

**Data Availability Statement:** All relevant data are within the paper and its Supporting Information files.

## Abstract

### Objective

Airway inflammation in asthma involves not only the central airways but extends to peripheral airways. Lung deposition may be key for an appropriate treatment of asthma. We compared the clinical effects of extrafine hydrofluoroalkane (HFA)-beclomethasone-formoterol (BDP-F) versus equipotent doses of nonextrafine combination of an inhaled corticosteroid and a long acting β2-agonist (ICS-LABA) in asthma.

### Methods

We identified eligible studies by a comprehensive literature search of PubMed, EMBASE and the Cochrane Central Register of Controlled Trials (CENTRAL). Data analysis was performed with the Review Manager 5.3.5 software (Cochrane IMS, 2014).

### Results

A total of 2326 patients with asthma from ten published randomized controlled trials (RCTs) were enrolled for analysis. Change from baseline in morning pre-dose peak expiratory flow (PEF), evening pre-dose PEF and forced expiratory volume in one second (FEV$_1$) were detected no significant differences between extrafine HFA-BDP-F and nonextrafine ICS-LABAs ($p = 0.23$, $p = 0.99$ and $p = 0.23$, respectively). Extrafine HFA-BDP-F did not show any greater benefit in forced expiratory flow between 25% and 75% of forced vital capacity (FEF25-75%), the parameter concerning peripheral airways (MD 0.03L/s, $p = 0.65$; n = 877). There were no substantial differences between interventions in fractional exhaled nitric oxide (FeNO) levels or in its alveolar fraction. The overall analysis showed no significant benefit of extrafine HFA-BDP-F over nonextrafine ICS-LABA in improving Asthma Control Test (ACT) score ($p = 0.30$) or decreasing the number of puffs of rescue medication use

**Funding:** The author(s) received no specific funding for this work.

**Competing interests:** The authors have declared that no competing interests exist.

($p$ = 0.16). Extrafine HFA-BDP-F did not lead to less exacerbations than nonextrafine ICS-LABA (RR 0.61, 95% CI: 0.31 to 1.20; $I^2$ = 0; $p$ = 0.15).

## Conclusion

Enrolled RCTs of extrafine HFA-BDP-F have demonstrated no significant advantages over the equivalent combination of nonextrafine ICS-LABA in improving pulmonary function concerning central airways or peripheral airways, improving asthma symptom control or reducing exacerbation rate.

## Introduction

Asthma is a heterogeneous disease leading to chronic inflammation in the airways. The combination treatment of an inhaled corticosteroid (ICS) and a long acting β2-agonist (LABA) is recommended as a first-line therapy for subjects with moderate-to-severe asthma [1]. Currently, four fixed dose combinations (FDCs) of ICS-LABA have been developed and commercially available [2]: fluticasone propionate-formoterol fumarate (FP-F), fluticasone proprionate-salmeterol xinafoate (FP-S), budesonide-formoterol fumarate (BUD-F) and beclometasone dipropionate-formoterol fumarate (BDP-F). Among current therapies, BDP-F fixed combination delivers extrafine particles of BDP (100 ug) and F (6 ug) per actuation via a hydrofluoroalkane (HFA) pressurized metered dose inhaler (pMDI), allowing high lung deposition and homogeneous distribution throughout the whole bronchial tree [3, 4].

As is known, airway inflammation in asthma involves not only the central airways but extends to peripheral airways [5]. The inflammatory process in the distal airways is similar to that in the central ones, but sometimes more severe, involving the adventitia more than submucosa [6]. Small airways (< 2 mm in diameter), significantly contribute to the pathogenesis of asthma in terms of bronchoconstriction and hyper-responsiveness [5, 7]. Increasing evidence is linked to a single reference from over 10 years ago in the correlation between small airways impairment and poor asthma control [8]. Moreover, the outer wall of small airways was found to be the major site of airway remodeling in patients with fatal asthma [9]. Inflammatory and functional changes of the small airways strongly contribute to the heterogeneous manifestations of chronic diseases such as asthma, suggesting that the site should not be neglected in the monitoring or management for the diseases [10]. These twenty years, various drug formulations and inhalation techniques have been developed in order to optimize the delivery of ICS-LABA to the whole bronchial tree.

Modulite® (Chiesi Farmaceutici, Parma, Italy), a new technology has been developed to obtain extrafine formulation of new drugs and reformulation of preexisting drugs. The particle size can be tailored within the respirable range (coarser particles or extrafine ones) by Modulite® technology and this property may lead to deeper pulmonary penetration [11, 12]. This technology has been applied to develop the first FDCs containing extrafine formulation of BDP and BDP-F both in the pMDI and dry powder inhaler (DPI) versions, with a mass median aerodynamic diameter (MMAD) of BDP 1.4 μm and FF 1.5 μm in pMDI and BDP 1.5 μm and FF 1.4 μm in NEXThaler, respectively [13]. Extrafine BDP-F of smaller particles is able to improve a more peripheral lung deposition of the medication [14] and distribute uniformly in the bronchial tree [15], compared to those of FP-S DPI and BUD-F DPI. In addition, the new therapeutic extrafine formulations inducing a more homogeneous drug distribution in the respiratory tract have been shown to reach the small airways with reduced oropharyngeal deposition [11].

Extrafine HFA-BDP has been found to be more potent than the nonextrafine formulation on the small airway inflammation due to its high penetration of the medication into the lung [16, 17]. A pilot study evaluated effects of treatment with extrafine BDP-F on airway function and the findings suggest that extrafine BDP-F has additional benefits on both large and small airways compared with nonextrafine FP-S [18]. A significant improvement of functional parameters reflecting small airway obstruction was also reported in another preliminary study [19]. In a randomized controlled parallel group trial comparing extrafine BDP-F and the marketed combinations of nonextrafine ICS-LABA, a significant decrease in fractional exhaled nitric oxide (FeNO) levels were observed in extrafine BDP-F, indicating an additional favorable effect on peripheral airway inflammation [20]. The reported adverse events of extrafine BDP-F were comparable to nonextrafine ICS-LABA in asthmatics [21–25]. Nevertheless, Papi et al found that the FDCs of BDP-F was not superior to nonextrafine ICS-LABA (FP-S or BUD-F) in terms of lung function, asthma control, symptom scores, rescue medication use or the rate of asthma exacerbations [23–25], except for the advantage of a more rapid onset of bronchodilation over FP-S [24].

Thus, although this new relevance appears to warrant a pharmacological advantage, its potency on clinical benefits in asthma needs to be confirmed. This study was designed to evaluate and compare the clinical effects and tolerability profile of treatments with extrafine HFA-BDP-F versus equipotent doses of nonextrafine ICS-LABA in patients with moderate to severe persistent asthma.

## Methods

This systematic review and meta-analysis was performed in accordance with Preferred Reporting Items for Systematic Reviews and Meta-Analyses guidance [26].

### Search strategy

We identified eligible studies by a comprehensive literature search of PubMed, Embase and Cochrane Library, using the combination of keywords including "beclomethasone" and "asthma". The last search was performed on June 2021. Detailed search strategies were shown in S1 File. Additional relevant and eligible studies were captured by scanning the reference lists of the included articles and previous reviews to minimize potential publication bias. We conducted a search of relevant files on the website of the international registry of clinical trials (http://www.clinicaltrials.gov/). We did not impose any restrictions on language of publication.

### Selection and exclusion criteria

Eligible studies were defined based on the following criteria: (1) Study design: we limited the specific publication type to randomized controlled trials (RCTs). We only included RCTs in this review because of their highest quality, in order to ensure the high quality of our data analysis. (2) Population: trials involving patients with a diagnosis of asthma regardless of age, gender or disease severity were included. (3) Comparison: intervention treatment was extrafine-particle ICS-LABA versus nonextrafine-particle ICS-LABA. (4) Outcomes: we included studies reporting any of the following end points: Measures of lung function: change from baseline in morning pre-dose peak expiratory flow (PEF, L/min), evening pre-dose PEF (L/min), forced expiratory volume in one second (FEV1, L), forced vital capacity (FVC, L), and other outcome variables of small airway dysfunction such as forced expiratory flow between 25% and 75% of forced vital capacity (FEF25-75%, L/s), Delta R5-R20 [kPa/(L*s)]; Airway inflammation: FeNO (ppb) and alveolar nitric oxide ($C_{Alv}$, ppb); Asthma control: Asthma Control Test (ACT) score, daytime symptom score and night-time symptom score, number of puffs of rescue medication

per day; Exacerbations: number of patients with at least 1 severe asthma exacerbation requiring administration of oral corticosteroids; Adverse events (AEs): incidence of any AEs and serious AEs. Serious AEs is defined as an event that jeopardizes the patient or may require medical or surgical intervention, such as fatal or life-threatening, results in persistent or significant disability/incapacity, requires inpatient hospitalization or prolongation of existing hospitalization, etc. Given their lacking information concerning essential data, trials published without full text such as conference papers solely in abstract form or letters were also excluded.

## Study selection, data extraction and assessment of risk of bias

Two reviewers worked independently in stages including study selection, data extraction and risk of bias assessment. All divergences were resolved by the involvement of a third reviewer to reach a mutual consensus.

The title and abstract of each study identified for inclusion was assessed for further full-text evaluation. We carried out the initial search reports or publications, identified studies according to the inclusion criteria and recorded ineligible studies with reasons for exclusion. Duplicates and collated multiple reports of the same study were discarded, so that each trial instead of each report was the unit of interest in this review.

Relative data from the included studies were separately extracted in a standardized Cochrane data extraction regulation [27]. In addition, we tried to search supplementary appendix or contact the corresponding author to acquire essential missing data.

The assessment of risk of biases was performed based on the methodology reported by each enrolled study following the rules specified in the Cochrane Handbook for Systematic Reviews of Interventions which consists of seven items in six domains [28]. The certainty of evidence was evaluated using the Grading of Recommendations Assessment, Development and Evaluation (GRAED) approach [29].

## Statistical analysis

Data analysis was performed by two independent statisticians blinded to the study protocol with the Review Manager 5.3.5 software (Cochrane IMS, 2014). We conducted analysis by intention to treat (ITT) involving all participants to minimize bias.

In order to include multiple interventions of each study and avoid double-counts of the patients in the shared intervention groups, doses were combined in only one group [27]. Dichotomous data and continuous data were analyzed as risk ratios (RRs) and mean differences (MDs) respectively, and all data were presented with 95% confidence intervals (CIs). Trials were pooled with a fixed-effect model, or a random-effect model if moderate or substantial heterogeneity was identified. We efficiently evaluated statistical heterogeneity by the $I^2$ test, with 25%, 26%-39%, 40%-60%, and 60%-100%, considered to represent absence, unimportant, moderate, and substantial degrees of inconsistency respectively [30]. $P$ values of $< 0.05$ (2-tailed test) were rendered statistically significant. Sensitivity analyses for some outcomes were also performed to replace alternative decisions or bounds of values for decisions that were subjective. We tried to make the sub-group analysis of treatment duration on effect of extrafine HFA-BDP-F versus nonextrafine ICS-LABA on AEs and serious AEs.

We failed to evaluate publication bias with a funnel plot and Egger's test, due to the limited number of included studies.

## Patient and public involvement

Patients and the public were not involved in the design and conception of this study. No patients were involved in developing the research question or the outcome measures. Our

study used aggregated data from already published researches, it is not easy to disseminate the results of the research to participants directly.

## Results

### Study identification and selection

The literature search process and study selection flow are shown in Fig 1. A more detailed description of the key studies excluded was shown in S2 File. We avoided double-counting of participants from overlapping studies. After thorough review, a total of 2326 patients with asthma from ten published RCTs were enrolled for analysis. All studies were published from 2007 to 2018 [18–25, 31, 32].

### Study characteristics

Additional detailed characteristics on the enrolled studies are summarized in Table 1. All studies were conducted in patients aged 18–65 years who had been diagnosed with asthma. The severity of disease was generally moderate to severe and baseline $FEV_1$ (% pre) ranged from 65.4±10.8 to 97.2±13.6 in the studies. All enrolled studies were randomized controlled trials of which two were crossover trials [19, 22] and eight studies were parallel-group trials [18, 20, 21, 23–25, 31, 32]. Treatment duration of studies ranged from 3 to 24 weeks. Patients were randomized to either extrafine HFA-BDP-F via pMDI (FOSTER[R]; Chiesi Farmaceutici, Italy) [18–25, 31, 32] or nonextrafine ICS-LABA, including FP-S via DPI (Seretide[R] Diskus[R]; GlaxoSmithKline, UK) [18–20, 23, 32], FP/S via pMDI (Seretide[R]; GlaxoSmithKline, UK) [21, 24], BUD-F via DPI (Symbicort[R] Turbuhaler[R]; AstraZeneca, Sweden) [20, 22, 25] and

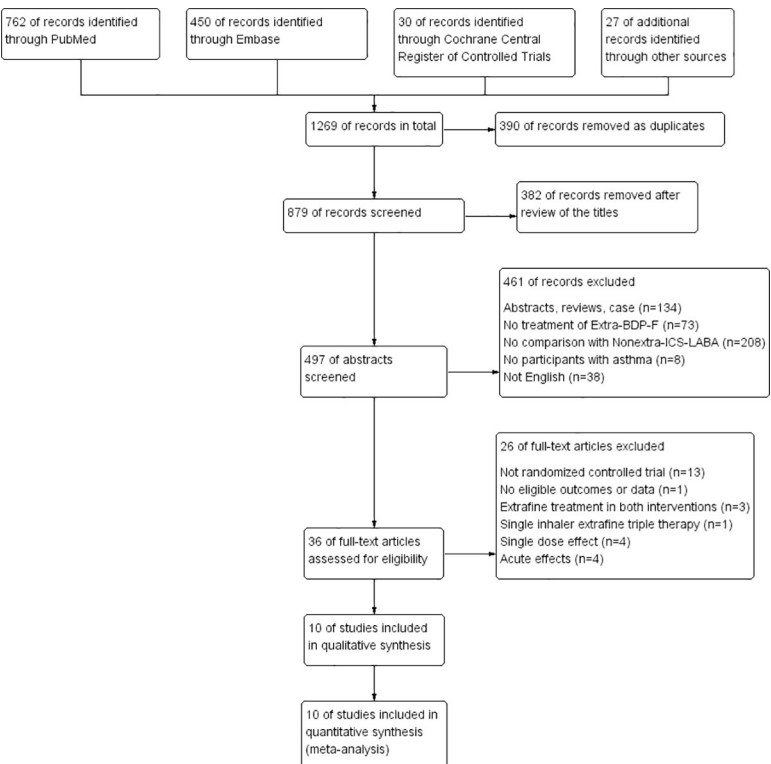

**Fig 1. Flowchart for identification of studies selected.**

Chloroflurocarbon (CFC)-BDP and formoterol via separate inhalers (Becloforte®; Allen & Hanburys, and Foradil® Aerolizer®; Novartis) [31]. All intervention medications were administered twice daily. Dosages and administrations are summarized in Table 1.

## Quality assessment

The risk of bias is presented briefly in Fig 2. On the whole, the included trials in our meta-analysis had relatively high quality. Blindness breaking were considered due to different inhalers used between the intervention groups, thus several studies were labeled with "high" risk of performance bias [18–20, 22, 23, 25, 31].

## Heterogeneit

There were no significant inconsistencies in almost all the outcomes. Pooled analysis of change of lung function such as morning and evening PEF, $FEV_1$, FVC and FEF25-75% were consistent ($I^2$ = 0); similar results were found in ACT scores, rescue medication use, incidence of exacerbations and adverse events; and unimportant heterogeneity existed in the change of daytime symptom score.

## Outcomes

**Lung function.** In general, extrafine HFA-BDP-F via pMDI showed no statistically significant improvement in pulmonary function compared with nonextrafine ICS-LABA. Change from baseline in morning pre-dose PEF or evening pre-dose PEF were detected no significant difference between interventions (MD -4.85 L/min, 95% CI: -12.72 to 3.02, $p$ = 0.23, n = 1535, Fig 3; MD -0.05 L/min, 95% CI: -7.38 to 7.27, $p$ = 0.99, n = 1119, S1 Fig). No significant increase in $FEV_1$ and FVC was found in patients receiving extrafine HFA-BDP-F compared to those receiving nonextrafine ICS-LABA (MD 0.03 L, $p$ = 0.23 n = 1565, Fig 4; MD 0.04 L, $p$ = 0.23 n = 1535, S2 Fig). The GRAED quality of morning or evening PEF, FEV1 and FVC were judged to be moderate (Table 2). In addition, pooled analysis did not show any greater benefit of extrafine HFA-BDP-F in the parameters concerning peripheral airways FEF25-75% (MD 0.03L/s, $p$ = 0.65; n = 877) and its percentage of predicted normal (MD 0.44, $p$ = 0.86; n = 526). No significant changes of other parameters concerning small airway dysfunction in body plethysmography, impulse oscillometry or forced spirometry were detected.

**Airway inflammation.** Due to incomplete data, we failed to conduct a pooled analysis of changes in airway inflammation. In Bulac 2015, there were no substantial changes in FeNO levels in nonextrafine FP-S or BUD-F but a significant decrease of 2.24 ppb in extrafine HFA-BDP-F ($p$ = 0.001) [20]. However, in another study, no significant decreases of FeNO from baseline were observed in either extrafine HFA-BDP-F or nonextrafine BUD-F [22]. Decreased mean $C_{Alv}$ were observed after 4 weeks of treatment and the adjusted geometric mean (log transformed data) was 0.942 ppb (95% CI: 0.778 to 1.141 ppb) with extrafine HFA-BDP-F and 0.903 ppb (95% CI:0.741–1.099 ppb) with nonextrafine BUD-F [22].

**Asthma control.** In brief, the overall analysis showed no significant benefit of extrafine HFA-BDP-F over nonextrafine ICS-LABA in asthma control. Change of ACT score from baseline was assessed in two studies [19, 21] and the pooled result was not significantly different between extrafine HFA-BDP-F group and the control group (MD -0.48, 95% CI: -1.39 to 0.43; $I^2$ = 0%; $p$ = 0.30). Similar findings were found in daytime symptom score change (MD 0.04, 95% CI: -0.09 to 0.18; $I^2$ = 38%; $p$ = 0.55) and night-time symptom score change (MD -0.02, 95% CI: -0.16 to 0.11; $I^2$ = 0; $p$ = 0.74).

**Rescue medication use.** There was no significant benefit of extrafine HFA-BDP-F over nonextrafine ICS-LABA in decreasing the number of puffs of daytime and night-time rescue

**Table 1. Characteristics of included studies.**

| Study | NCT No. | Study Design | Funding Sources | Age, Mean (SD), y | Age range, y | No. of Participants | No. of Female (%) | Baseline FEV$_1$, L (SD) | Baseline FEV$_1$, % pre (SD) | Severity |
|---|---|---|---|---|---|---|---|---|---|---|
| Barnes 2013 | NCT00901368 | multicentre, double-blind, double-dummy, randomized, parallel group, controlled clinical study | Chiesi Farmaceutici SpA, Italy | 44.0 (13.9) | 18–65 | 416 | 235 (56.5) | 3.13 (0.82) | 97.2 (13.6) | - |
| Bulac 2015 | - | randomized, parallel group, controlled clinical trial | No | 42.6 (12.3) | 18–65 | 95 | 80 (84.2) | - | 86.4 (10.6) | - |
| Corda 2011 | NCT01255579 | double-blind, double-dummy randomized crossover study | Unknown | 45.0 (10.0) | >18 | 8 | 5 (62.5) | - | 85.0 (17.9) | moderate |
| Hsieh 2017 | - | phase III, multicentre, double-blind, double dummy, two-arm parallel, randomized study | Orient EuroPharma Co., Ltd, Taiwan | 45.1 (14.5) | 20–65 | 244 | 117 (48.0) | 2.20 (0.75) | - | moderate-to-severe |
| Huchon 2009 | NCT00476268 | phase III, multicentre, multinational, double-blind, double-dummy, clinical trial | Chiesi Farmaceutici SpA, Italy | 47.4 (12.5) | 18–70 | 645 | 415 (64.3) | 1.97 (0.55) | 65.4 (10.8) | moderate-to-severe |
| Kirsten 2015 | - | phase IV, single-centre, double-blind, double-dummy, two-way cross-over, randomised study | Chiesi Farmaceutici SpA, Italy | 37.4 (9.0) | 18–65 | 22 | 13 (59.1) | 3.54 (0.85) | 91.6 (13.0) | mild-to-moderate |
| Papi 2007 (1) | NCT00394368 | phase III, multicentre, multinational, double-blind, randomized, two-arm parallel group, controlled study | Chiesi Farmaceutici SpA, Italy | 48.5 (11.5) | 18–65 | 228 | 128 (56.1) | 2.03 (0.51) | 67.3 (9.6) | moderate-to-severe |
| Papi 2007 (2) | - | phase III, multicentre, multinational, double-blind, double-dummy, randomised, two-arm parallel group, controlled study | Chiesi Farmaceutici SpA, Italy | 44.7 (11.8) | 18–65 | 216 | 125 (57.9) | 2.25 (0.68) | 69.9 (10.2) | moderate-to-severe |
| Papi 2012 | NCT00497237 | randomised, two-arm parallel group, controlled, study | Chiesi Farmaceutici SpA, Italy | 44.0 (13.0) | 18–65 | 422 | 276 (65.4) | 2.95 (0.85) | 87.6 (14.6) | - |
| Scichilone 2010 | - | double-blind, double dummy, randomized, parallel group study | Chiesi Farmaceutici SpA, Italy | 43.0 (11.8) | 18–50 | 30 | 15 (50.0) | 2.26 (0.74) | 71.1 (13.7) | moderate |

| Study | Experimental Interventions | Control Interventions | Duration, weeks | Primary Outcomes |
|---|---|---|---|---|
| Barnes 2013 | HFA-BDP-F (pMDI) 200/12ug bid | FP-S (DPI) 250/50ug bid | 12 | pre-dose morning FEV$_1$ (L) at the end of the 12-week treatment period |
| Bulac 2015 | HFA-BDP-F (pMDI) 100/6ug bid | FP-S (DPI) 500/50ug bid; BUD-F (DPI) 320/9ug bid | 3 | spirometry, exhaled nitric oxide (eNO) levels, and small airway functional indices, namely, Sacin and Scond values |
| Corda 2011 | HFA-BDP-F (pMDI) 200/12ug bid | FP-S (DPI) 250/50ug bid | 24 | Asthma Control Test, pulmonary function testing |
| Hsieh 2017 | HFA-BDP-F (pMDI) 200/12ug bid | FP-S (pMDI) 250/50ug bid | 12 | the mean change from pre-dose FEV$_1$ on week 0–5 min post-dose FEV$_1$ on week 12 |
| Huchon 2009 | HFA-BDP-F (pMDI) 200/12ug bid + Placebo (DPI) bid + Placebo (pMDI) bid | CFC-BDP (pMDI) 500ug bid + F (DPI) 12ug bid + Placebo (pMDI) bid; CFC-BDP (pMDI) 500ug bid + Placebo (DPI) bid + Placebo (pMDI) bid | 24 | mean morning peak expiratory flow (PEF) measured by the patient before dosing during the last 14 days of treatment |
| Kirsten 2015 | HFA-BDP-F (pMDI) 100/6ug bid | BUD-F (DPI) 160/4.5ug bid | 4 | alveolar nitric oxide concentrations and peripheral lung resistance measured by forced oscillation |
| Papi 2007 (1) | HFA-BDP-F (pMDI) 200/12ug bid | FP-S (pMDI) 250/50ug bid | 12 | morning pre-dose PEF measured by patients in the last 2 weeks of treatment period |
| Papi 2007 (2) | HFA-BDP-F (pMDI) 200/12ug bid | BUD-F (DPI) 320/9ug bid | 12 | morning pre-dose PEF measured by patients, at least 12 h after the previous evening dose, in the last 2 weeks of the treatment period |
| Papi 2012 | HFA-BDP-F (pMDI) 200/12ug bid | FP-S (DPI) 250/50ug bid | 24 | the change in morning peak expiratory flow (PEF) values between baseline and the end of treatment |
| Scichilone 2010 | HFA-BDP-F (pMDI) 200/12ug bid | FP-S (DPI) 250/50ug bid | 12 | Methacholine (Mch) bronchoprovocation challenge and single breath nitrogen (sbN2) test |

HFA: hydrofluoroalkane; pMDI: pressurized metered dose inhaler; BDP: beclometasone dipropionate; FP: fluticasone propionate; S: salmeterol xinafoate; BUD: budesonide; F: formoterol fumarate; PEF: peak expiratory flow; FEV$_1$: forced expiratory volume in one second; eNO: exhaled nitric oxide; Mch: methacholine; sbN2: single breath nitroge

**Table 2. Summary of findings including GRADE quality assessment of evidence trials.**

| No. of studies | Design | Risk of bias | Inconsistency | Indirectness | Imprecision | Other considerations | Extrafine BDP-F | Nonextrafine ICS-LABA | Relative (95%CI) | Absolute | Quality | Importance |
|---|---|---|---|---|---|---|---|---|---|---|---|---|
| **Quality assessment** | | | | | | | **No. of patients** | | **Effect** | | | |
| AEs | | | | | | | | | | | | |
| 7 | randomized trials | serious [1,2] | no serious inconsistency | no serious indirectness | serious [3] | none | 266/1002 (26.5%) | 270/1020 (26.5%) | RR 1.01 (0.89 to 1.14) | 3 more per 1000 (from 29 fewer to 37 more) | ⊕⊕⊕◯ LOW | CRITICAL |
| | | | | | | | | 16.5% | | 2 more per 1000 (from 18 fewer to 23 more) | | |
| AEs-Treatment more than 12 weeks | | | | | | | | | | | | |
| 6 | randomized trials | serious [1,2] | no serious inconsistency | no serious indirectness | no serious imprecision | none | 258/980 (26.3%) | 262/998 (26.3%) | RR 1.01 (0.89 to 1.14) | 3 more per 1000 (from 29 fewer to 37 more) | ⊕⊕⊕⊕ MODERATE | CRITICAL |
| | | | | | | | | 15.4% | | 2 more per 1000 (from 17 fewer to 22 more) | | |
| AEs-Treatment less than 12 weeks | | | | | | | | | | | | |
| 1 | randomized trials | serious | no serious inconsistency | no serious indirectness | serious [3] | none | 8/22 (36.4%) | 8/22 (36.4%) | RR 1 (0.46 to 2.19) | 0 fewer per 1000 (from 196 fewer to 433 more) | ⊕⊕◯◯ LOW | CRITICAL |
| | | | | | | | | 36.4% | | 0 fewer per 1000 (from 197 fewer to 433 more) | | |
| Serious AEs | | | | | | | | | | | | |
| 7 | randomized trials | serious [1,2] | no serious inconsistency | no serious indirectness | serious [3] | none | 7/1002 (0.7%) | 9/1020 (0.9%) | RR 0.8 (0.31 to 2.07) | 2 fewer per 1000 (from 6 fewer to 9 more) | ⊕⊕◯◯ LOW | IMPORTANT |
| | | | | | | | | 0% | | - | | |
| Serious AEs-Treatment more than 12 weeks | | | | | | | | | | | | |
| 6 | Randomized trials | serious [1,2] | no serious inconsistency | no serious indirectness | no serious imprecision | none | 7/980 (0.7%) | 9/998 (0.9%) | RR 0.8 (0.31 to 2.07) | 2 fewer per 1000 (from 6 fewer to 10 more) | ⊕⊕⊕⊕ MODERATE | IMPORTANT |
| | | | | | | | | 0.5% | | 1 fewer per 1000 (from 3 fewer to 5 more) | | |
| Serious AEs-Treatment less than 12 weeks | | | | | | | | | | | | |
| 1 | Randomized trials | serious [2] | no serious inconsistency | no serious indirectness | serious [3] | none | 0/22 (0%) | 0/22 (0%) | not pooled | not pooled | ⊕⊕◯◯ LOW | IMPORTANT |
| | | | | | | | | 0% | | not pooled | | |

(*Continued*)

**Table 2.** (Continued)

| No. of studies | Design | Quality assessment | | | | | No. of patients | | Effect | | Quality | Importance |
|---|---|---|---|---|---|---|---|---|---|---|---|---|
| | | Risk of bias | Inconsistency | Indirectness | Imprecision | Other considerations | Extrafine BDP-F | Nonextrafine ICS-LABA | Relative (95%CI) | Absolute | | |
| **FEV1(L)** | | | | | | | | | | | | |
| 6 | Randomized trials | serious [1,2] | no serious inconsistency | no serious indirectness | no serious imprecision | none | 774 | 791 | - | MD 0.03 higher (0.02 lower to 0.08 higher) | ⊕⊕⊕O MODERATE | CRITICAL |
| **FVC(L)** | | | | | | | | | | | | |
| 5 | Randomized trials | serious [1,2] | no serious inconsistency | no serious indirectness | no serious imprecision | none | 759 | 776 | - | MD 0.04 higher (0.03 lower to 0.11 higher) | ⊕⊕⊕O MODERATE | CRITICAL |
| **MMEF (L/s)** | | | | | | | | | | | | |
| 3 | Randomized trials | serious [1,2] | serious | no serious indirectness | no serious imprecision | none | 433 | 444 | - | MD 0.03 higher (0.11 lower to 0.17 higher) | ⊕⊕OO LOW | IMPORTANT |
| **MMEF(% pre)** | | | | | | | | | | | | |
| 2 | Randomized trials | serious [1,2] | no serious inconsistency | no serious indirectness | serious [3] | none | 250 | 276 | - | MD 0.44 higher (4.43 lower to 5.32 higher) | ⊕⊕OO LOW | IMPORTANT |
| **ACT** | | | | | | | | | | | | |
| 2 | randomized trials | serious [1,2] | no serious inconsistency | no serious indirectness | serious [3] | none | 127 | 133 | - | MD 0.48 lower (1.39 lower to 0.43 higher) | ⊕⊕OO LOW | IMPORTANT |
| **Day-time symptom score** | | | | | | | | | | | | |
| 2 | randomized trials | serious [1,2] | no serious inconsistency | no serious indirectness | serious [3] | none | 222 | 222 | - | MD 0.04 higher (0.09 lower to 0.18 higher) | ⊕⊕OO LOW | IMPORTANT |
| **Night-time symptom score** | | | | | | | | | | | | |
| 2 | randomized trials | serious [1,2] | no serious inconsistency | no serious indirectness | serious [3] | none | 222 | 222 | - | MD 0.02 lower (0.16 lower to 0.11 higher) | ⊕⊕OO LOW | IMPORTANT |
| **Rescue medication use (puffs/day)** | | | | | | | | | | | | |
| 3 | randomized trials | serious [1,2] | no serious inconsistency | no serious indirectness | no serious imprecision | none | 433 | 442 | - | MD 0.1 higher (0.04 lower to 0.25 higher) | ⊕⊕⊕O MODERATE | IMPORTANT |
| **Exacerbations** | | | | | | | | | | | | |

*(Continued)*

**Table 2.** (Continued)

| No. of studies | Design | Risk of bias | Inconsistency | Indirectness | Imprecision | Other considerations | Extrafine BDP-F | Nonextrafine ICS-LABA | Relative (95%CI) | Absolute | Quality | Importance |
|---|---|---|---|---|---|---|---|---|---|---|---|---|
| 6 | randomized trials | serious [1,2] | no serious inconsistency | no serious indirectness | no serious imprecision | none | 13/973 (1.3%) | 22/993 (2.2%) | RR 0.61 (0.31 to 1.2) | 9 fewer per 1000 (from 15 fewer to 4 more) | ⊕⊕⊕◯ MODERATE | CRITICAL |
| | | | | | | | | 1.9% | | 7 fewer per 1000 (from 13 fewer to 4 more) | | |
| **Morning PEF (L/min)** | | | | | | | | | | | | |
| 5 | randomized trials | serious [1,2] | no serious inconsistency | no serious indirectness | no serious imprecision | none | 759 | 776 | - | MD 4.85 lower (12.72 lower to 3.02 higher) | ⊕⊕⊕◯ MODERATE | CRITICAL |
| **Evening PEF (L/min)** | | | | | | | | | | | | |
| 4 | randomized trials | serious [1,2] | no serous inconsistency | no serious indirectness | no serous imprecision | none | 552 | 567 | - | MD 0.05 lower (7.38 lower to 7.27 higher) | ⊕⊕⊕◯ MODERATE | CRITICAL |

GRADE: Grading of Recommendations Assessment, Development and Evaluation; AEs: adverse events; RR: risk ratios; MD: mean differences; FEV₁: forced expiratory volume in one second; FVC: forced vital capacity; MMEF: maximum midexpiratory flow; ACT: Asthma Control Test; PEF: peak expiratory flow.

[1] Some of the trials did not state the randomized method and concealment of allocation.

[2] Some of the trials did not blind the participants and personnel.

[3] Small number of patients.

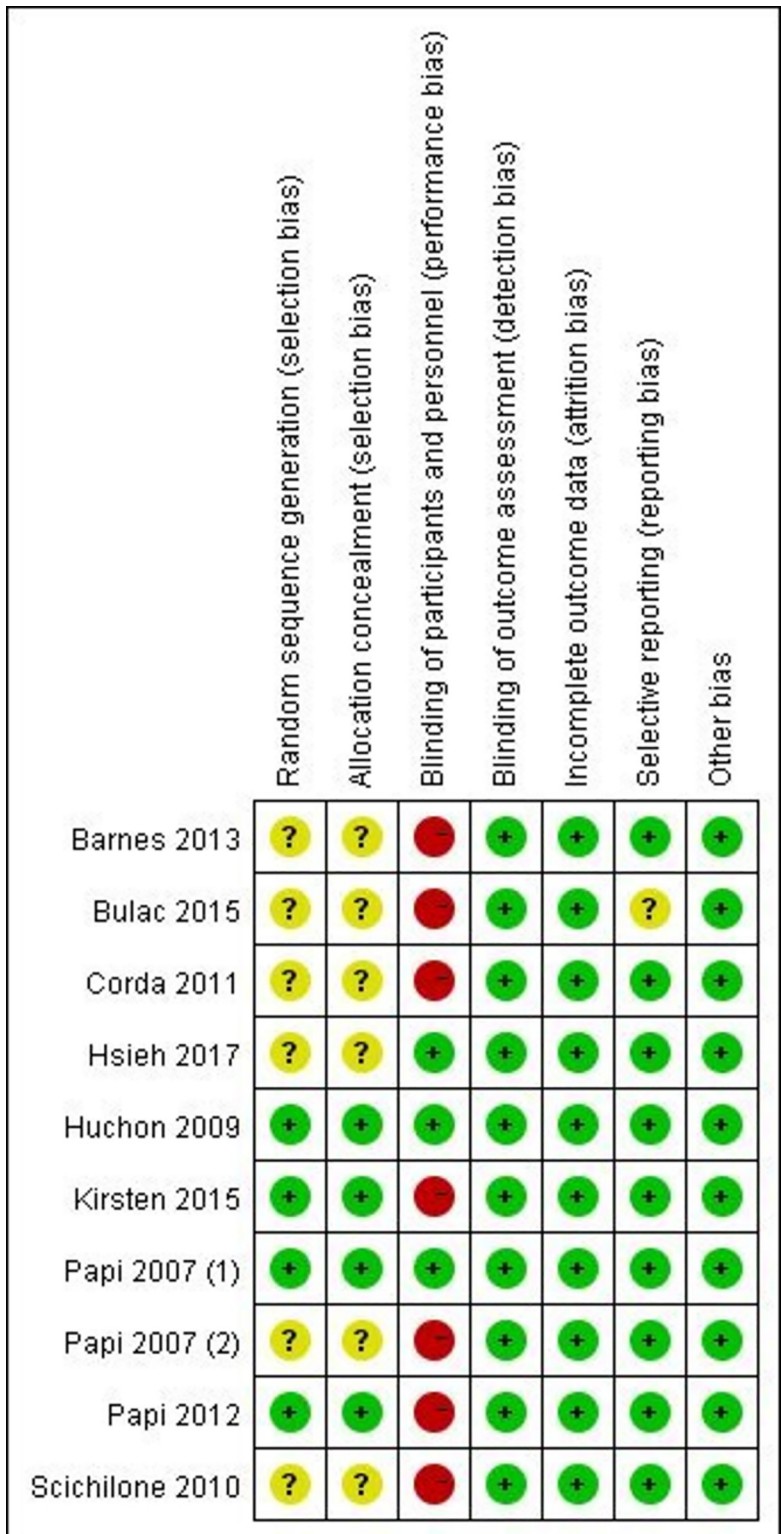

**Fig 2. Risk of bias summary of included studies.**

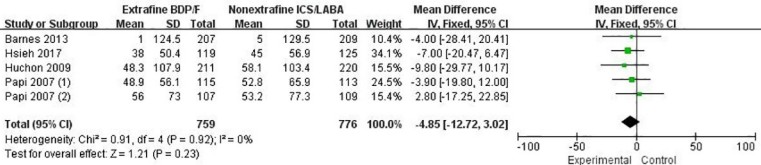

**Fig 3. The effect of extrafine HFA-BDP-F versus nonextrafine ICS-LABA on morning pre-dose peak expiratory flow change from baseline.**

medication use in pooled data of three trials [24, 25, 31] (MD 0.10, 95% CI: -0.04 to 0.25; $I^2$ = 0; $p$ = 0.16).

**Exacerbations.** Data of exacerbation rate was measured in six trials with 1966 patients [21, 23–25, 31, 32] and the pooled effect showed that extrafine HFA-BDP-F might lead to less exacerbations than nonextrafine ICS-LABA but the difference was not statistically significant (RR 0.61, 95% CI: 0.31 to 1.20; $I^2$ = 0; $p$ = 0.15; Fig 5). Certainty in the evidence was judged to be moderate, mainly because of not blinding the participants and personnel in some trials (Table 2). In addition, the time to the first exacerbation in the Kaplan–Meier survival estimate was not statistically different between groups, as reported in three studies ($p$ = 0.358, 0.342, 0.36, respectively) [23–25].

**Adverse events.** The incidence of any AEs and serious AEs between extrafine HFA-BDP-F and nonextrafine ICS-LABA were reported in 7 trials containing 2022 patients [21–25, 31, 32]. There were no meaningful differences observed in any AEs (RR 1.01; $I^2$ = 0; $p$ = 0.92; Fig 6) or serious AEs (RR 0.80; $I^2$ = 0; $p$ = 0.65; S3 Fig). But the evidences of AEs and serious AEs were judged to be low, mainly because of small number of patients in some trials (Table 2). The subgroup analysis showed no differences between a longer treatment duration of more than 12 weeks and less than 12 weeks (S4 and S5 Figs). No death was reported in patients receiving either intervention during the trials. Overall, all the studies having monitored vital signs (such as cardiac frequency and blood pressure), laboratory parameters and electrocardiogram (ECG) reported the treatment groups were well tolerated and no notable differences were detected between groups.

## Discussion

As we know, this was the first review and meta-analysis comparing extrafine BDP-F fixed combination via pMDI with other ICS-LABAs. In this review, we demonstrated that extrafine HFA-BDP-F was as effective as FP-S or BUD-F in maintaining asthma symptom control in patients with asthma as measured via the validated questionnaire, ACT score. Extrafine HFA-BDP-F showed no more benefits in improving pulmonary function concerning central airways or peripheral airways, or reducing airway inflammation, compared with nonextrafine ICS-LABAs.

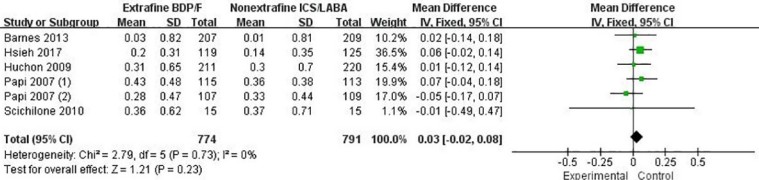

**Fig 4. The effect of extrafine HFA-BDP-F versus nonextrafine ICS-LABA on forced expiratory volume in one second.**

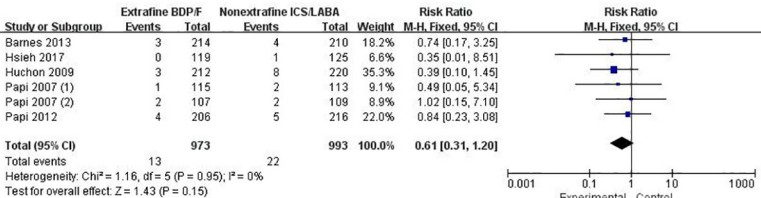

**Fig 5. The effect of extrafine HFA-BDP-F versus nonextrafine ICS-LABA on exacerbation rate.**

Both extrafine HFA-BDP-F and nonextrafine ICS-LABAs achieved comparable improvements for the endpoints as demonstrated by the increases in lung function during the study course in the pooled analysis, confirming that recruited moderate or severe asthmatics, unstable with the present treatment such as a daily dose of up to 1000 ug BDP or equivalent, were in need of step-up therapy with combination. Our study did not demonstrate superiority of extrafine HFA-BDP-F over nonextrafine FP-S or BUD-F in the mean change from baseline morning pre-dose PEF or evening pre-dose PEF between groups. The difference in corticosteroids dose between the treatment groups does not imply a difference in systemic exposure, since this depends not only on the nominal dose but also on the amount of drug reaching the lungs and pharmacokinetics properties of the corticosteroids [25].

As for airway inflammation, although some published studies suggested that specific formulations of ICS or ICS-LABA can modify biomarkers or parameters related to small airways, we failed to conduct a meta-analysis due to the limited number of included studies. One of the main issues in the evaluation of small airway function is the lack of a gold standard for the functional measurement of the distal lung. Techniques to test small airway function include sputum induction, impulse oscillometry, the nitrogen washout test, and alveolar fraction of exhaled nitric oxide derived by measurements of nitric oxide at multiple expiratory flows [10, 33]. Kirsten and colleagues collected induced sputum in their study but no significant changes regarding the sputum cellular composition were observed after treatment with extrafine HFA-BDP-F or nonextrafine BUD-F [22]. Delta R5-R20 in impulse oscillometry, as a parameter of small airway function, has been demonstrated to significantly be related to disease control in asthmatics [8]. A clinical trial comparing ciclesonide with fluticasone propionate in patients with mild asthma showed significant improvements in distal reactance (X5) and resistance of small airways (Delta R5-R20) [34]. Kirsten and colleagues also observed significant changes of Delta R5-R20 without significant changes of $FEV_1$, further strengthening the evidence that changes of small airway resistance following intervention can be demonstrated in the absence of $FEV_1$ improvements [22]. The characteristics of small airway obstruction include premature airway closure and air trapping, which can be assessed by an elevation of residual volume (RV) or RV/total lung capacity (TLC) ratio [35]. It is a pity that we were not

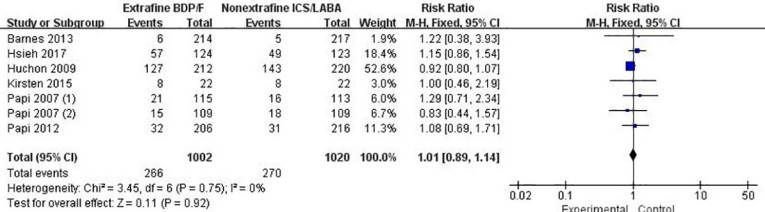

**Fig 6. The effect of extrafine HFA-BDP-F versus nonextrafine ICS-LABA on adverse events.**

able to analyses this parameter from the published RCTs. The significant decrease in FeNO levels or airway resistance in the small-particle combination group might attribute to the suppression of peripheral airway inflammation [20, 36]. Moreover, extrafine BDP has been previously reported with reduced air trapping compared with nonextrafine BDP and this has been hypothesized to be associated with greater peripheral airways patency due to more effective suppression of small airways inflammation [37]. No improvement in FVC with extrafine HFA-BDP-F reported in this study is not consistent with a greater reduction in small airways obstruction and air trapping in the previous studies [17, 38, 39].

It has been postulated that asthma control may be improved by targeting small airways. The real-life studies demonstrated the value of extrafine BDP-F in achieving asthma control [14] and improving quality of life [40]. Huchon et al [31] confirmed a significantly higher percentage of days with asthma control in the extrafine HFA-BDP-F than in the nonextrafine CFC-BDP and formoterol combination (least squares mean difference 7.66, 95% CI: 2.46 to 12.87; $p$ = 0.004). However, in our study, comparable improvements in both groups were observed in the assessment of clinical symptoms. ACT score, daytime symptom score, and night-time symptom score change improved from baseline by a similar degree in both treatment groups.

Frequency of rescue medication use and exacerbation rates were also similar with no statistically significant difference between groups. However, it was not possible to treat exacerbation rate as a primary end-point in the present study, since exposure time was limited and more patients are needed in order to detect potential differences between treatments.

BDP delivered via a pMDI is an established ICS available worldwide, which allows patients not adequately controlled with ICS alone to continue using the same device with the same molecule and the same inhalation technique. A limitation in this study is the lack of the relation between asthma control and future risk of exacerbation due to the limited exposure time. More patients and further long-term studies are needed in order to detect potential differences between treatments. Second, some patients with mild-to-moderate asthma were relatively well controlled with initial medication. Therefore, there might have been little room for improvements in the studies. It's a pity we failed to evaluate publication bias with a funnel plot and Egger's test, due to the limited studies in the meta-analysis. Moreover, in most trials, MDI and DPI were used in the experimental group and the control group, respectively, resulting in difficulty to implement blindness.

## Conclusion

As we know, our study is the first meta-analysis to compare the efficacy and safety of a pMDI containing beclomethasone and formoterol with a standard combination of BUD-F or FP-S in asthmatics whose symptoms were not adequately controlled with ICS alone. Enrolled RCTs of extrafine HFA-BDP-F have demonstrated no significant advantages over the equivalent combination of nonextrafine ICS-LABA in improving pulmonary function concerning central airways or peripheral airways, improving asthma symptom control or reducing exacerbation rate.

## Supporting information

**S1 File. Search strategy used to identify trials.**
(DOCX)

**S2 File. A detailed description of the key studies excluded.**
(DOCX)

**S1 Fig. The effect of extrafine HFA-BDP-F versus nonextrafine ICS-LABA on evening pre-dose peak expiratory flow change from baseline.**
(TIF)

**S2 Fig. The effect of extrafine HFA-BDP-F versus nonextrafine ICS-LABA on forced vital capacity change from baseline.**
(TIF)

**S3 Fig. The effect of extrafine HFA-BDP-F versus nonextrafine ICS-LABA on serious adverse events.**
(TIF)

**S4 Fig. The subgroup analysis of treatment duration on effect of extrafine HFA-BDP-F versus nonextrafine ICS-LABA on adverse events.**
(TIF)

**S5 Fig. The subgroup analysis of treatment duration on effect of extrafine HFA-BDP-F versus nonextrafine ICS-LABA on serious adverse events.**
(TIF)

**S1 Table. PRISMA 2009 checklist.**
(DOCX)

## Acknowledgments

We thank all authors who provided published information for our meta-analysis.

## Author Contributions

**Conceptualization:** Chuntao Liu.

**Data curation:** Ting Liu, Dan Yang.

**Formal analysis:** Ting Liu, Dan Yang.

**Investigation:** Ting Liu, Dan Yang.

**Methodology:** Chuntao Liu.

**Project administration:** Chuntao Liu.

**Resources:** Chuntao Liu.

**Software:** Ting Liu, Dan Yang.

**Supervision:** Chuntao Liu.

**Validation:** Ting Liu, Dan Yang.

**Visualization:** Ting Liu, Dan Yang.

**Writing – original draft:** Ting Liu, Dan Yang.

**Writing – review & editing:** Ting Liu, Dan Yang.

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
