## [Decision Letter · Decision Letter 0]

7 May 2021

PONE-D-20-35965

Extrafine HFA-beclomethasone-formoterol vs nonextrafine combination of an inhaled corticosteroid and a long acting β2-agonist in patients with persistent asthma: a systematic review and meta-analysis

PLOS ONE

Dear Dr. Liu,

Thank you for submitting your manuscript to PLOS ONE. It was challenging and time consuming to find experts who are available to review this particular manuscript. Over 15 invitations to review the article were not accepted. However, the review is complete now and, after careful consideration, we feel that the article has merit but does not fully meet PLOS ONE’s publication criteria as it currently stands. Please see below my and the reviewer's comments raised during the review process. We invite you to submit a revised version of the manuscript that addresses these.

We look forward to receiving your revised manuscript.

Kind regards,

Daoud Al-Badriyeh

Academic Editor

PLOS ONE

Additional Editor Comments:

- I suggest sub-group analyses to see the impact of dose (200/12ug bid versus 100/6ug bid) and the duration of therapy on outcomes. Lower doses and shorted durations may have reduced the rate of adverse events as example.

- Include the funding source behind the RCTs you included in the meta-analysis.

- It is customary to associate the quantitative results of a meta-analysis with a classification of clinical evidence and the strength of recommendations, such as via the GRADE assessment in meta-analyses, particularly as authors did not perform a trial sequential analysis in the meta-analysis.

- I did not find the registration number of this meta-analysis in a prospective register (as PROSPERO) in order to follow the pre-planned protocol.

- Please elaborate on why only including RCTs, when only less than 10 studies were found in the literature and these are mostly not recent.

- Please include a list of excluded full text articles as appendix, including reasons for exclusion.

- What were the serious adverse events that you pool analyzed?

Journal Requirements:

2) Please include your tables as part of your main manuscript and remove the individual files. Please note that supplementary tables (should remain/ be uploaded) as separate "supporting information" files

Reviewers' comments:

Reviewer's Responses to Questions

**Comments to the Author**

1. Is the manuscript technically sound, and do the data support the conclusions?

Reviewer #1: Yes

2. Has the statistical analysis been performed appropriately and rigorously? 

Reviewer #1: Yes

3. Have the authors made all data underlying the findings in their manuscript fully available?

Reviewer #1: Yes

4. Is the manuscript presented in an intelligible fashion and written in standard English?

Reviewer #1: Yes

5. Review Comments to the Author

Reviewer #1: Dear Correspond Author,

I comment you and your research team for conducting the systematic review and meta-analysis to answer a question most clinicians who manage asthma patients have. You also followed the best practices in conducting a systematic review and meta analysis.

Below are some comments/suggestions and questions I have for you.

1. The protocol of the study is not part of the appendix. Has it been published anywhere? Is it possible to include the protocol in the revision to check for adherence to the protocol and to ensure study can be replicated. The manuscript does not report on any protocol deviation.

2. The key words on the cover page (4 of them) are more than the key words listed in the search strategy. Is there any reason that key words used in the search are only 2? If there were more key words, please consider including them in the methods. How were you able to ensure that all the relevant articles were included with using only the 2 keys words? From fig 1 flow chart, 929 TOTAL articles were identified from the 3 databases. Is it possible to provide details about the total number of articles identified in each data base after key word search and how many were excluded after the review of the titles of these studies?

3. Line 45 of the abstract results "didn't"- Please consider "did not"

4. Line 74 of the introduction: "These years, various drug formulations and inhalation techniques have been developed

..." - It is not very clear what these years mean?

5. Line 80 of the introduction "pressurized metered dose inhaler (pMDI) " -Consider using only pMDI as the abbreviation

has been defined above already.

6. Line 83 of introduction "to generate" -Please consider another word. For the context of delivering medication to a

site of action, "generate" may not be the most appropriate word to use.

7. Line 126 of the methods "and other outcome variables of" - Please check the font size of these words.

They are smaller.

8. Line 147 & 148 of methods "The assessment of risk of biases was performed based on the methodology reported

by each enrolled study following the rules specified in the Cochrane Handbook " - Why should the risk of biases be

based on the methodology of each enrolled study when only RCTs were included? It may be good to elaborate a

little bit more.

9. Line 155 of the statistical analysis "group27 " - Please check if it is a spelling mistake.

10. Line 164 of the statistical analysis "We failed to evaluate publication bias with a funnel plot and Egger’s test, due

to the limited number of included studies. "- Should the inability to check for publication bias not be included as

a limitation of the study?

11. Line 213 of the results outcomes "bodyplethysmography" - Please check the spelling.

12. Discussion section: How will the fewer number of studies and by extension relatively lower number of patients

affect the results?

13. References section: Line 346, reference 1- Please consider including the date you accessed this reference via the

website.

14. Appendix: Titles of figures 3,4,& 5: These titles need to be revised slightly. From the way it is written, the respective outcomes have effects on the treatment.

Thank you.

6. PLOS authors have the option to publish the peer review history of their article (what does this mean?). If published, this will include your full peer review and any attached files.

Reviewer #1: No

---

## [Author Response · Author response to Decision Letter 0]

21 Jun 2021

Dear editor,

Thank you for your letter and for the reviewers’ comments concerning our manuscript entitled “Extrafine HFA-beclomethasone-formoterol vs nonextrafine combination of an inhaled corticosteroid and a long acting β2-agonist in patients with persistent asthma: a systematic review and meta-analysis” (ID: PONE-D-20-35965). Those comments are valuable and very helpful for revising and improving our paper. We have carefully revised the manuscript according to the reviewers' comments, and the amendments are highlighted in yellow in the revised manuscript. Point by point responses to the reviewers’ comments are as flowing.

Responses to editor 

1. I suggest sub-group analyses to see the impact of dose (200/12ug bid versus 100/6ug bid) and the duration of therapy on outcomes. Lower doses and shorted durations may have reduced the rate of adverse events as example.

Response: Dear editor, thank you very much for your kind help! Your suggestion is valuable and very helpful to us. Sub-group analyses were considered when we extracted the data of these trials, however, it’s a pity we did not show them in the manuscript because of the limited number of studies, for example, there was only two small-sample studies (Bulac 2015 and Kirsten 2015) of low dose (200/12ug bid) and short duration (3-4 weeks). According to your suggestion, we have added the sub-group analyses of AE and SAE in the S4 and S5 Figs. (Line 262, 474-477)

2. Include the funding source behind the RCTs you included in the meta-analysis.

Response: Thank you for your reminder. We have added the funding sources in Table 1. (Page 10)

3. It is customary to associate the quantitative results of a meta-analysis with a classification of clinical evidence and the strength of recommendations, such as via the GRADE assessment in meta-analyses, particularly as authors did not perform a trial sequential analysis in the meta-analysis.

Response: According to the GRADE assessment system, RCT is of the highest quality level based on the research design. All we have enrolled were RCTs and we made a strict quality assessment according to the entry of PRISMA 2009 checklist which was included in the S1 Table and Fig 2. (Risk of bias summary). We considered the quality of these RCTs were relatively high except for the inevitable unblinding of MDI and DPI. (Line 478)

4. I did not find the registration number of this meta-analysis in a prospective register (as PROSPERO) in order to follow the pre-planned protocol.

Response: Thank you very much for your kind reminder. We have made a protocol for the meta-analysis, but we did not register in PROSPERO because of its website maintenance when we did this research. We have resubmit the protocol to PROSPERO. 

5. Please elaborate on why only including RCTs, when only less than 10 studies were found in the literature and these are mostly not recent.

Response: We only included RCTs because of their highest quality, in order to ensure the high quality of our data analysis. We searched several databases and the results recently, and these studies were indeed all RCT studies. Our research team has also carried out an RCT of BDP-F vs BUD-F, but the results have not been published.

6. Please include a list of excluded full text articles as appendix, including reasons for exclusion.

Response: We have made a more detailed description of the key studies excluded, which was shown in S2 File. ( Line 173-174, 467)

7. What were the serious adverse events that you pool analyzed?

Response: An SAE is not always the same in different studies, most of which is considered to meet any one of the following criteria: is fatal or life-threatening; results in persistent or significant disability/incapacity; constitutes a congenital anomaly/birth defect; requires inpatient hospitalization or prolongation of existing hospitalization; is medically significant, i.e. defined as an event that jeopardizes the patient or may require medical or surgical intervention. 

Journal Requirements

Response: We have revised the manuscript to meets PLOS ONE's style requirements.

Response: We have included our tables as part of our main manuscript and remove the individual files.

Responses to reviewer #1

1. The protocol of the study is not part of the appendix. Has it been published anywhere? Is it possible to include the protocol in the revision to check for adherence to the protocol and to ensure study can be replicated. The manuscript does not report on any protocol deviation.

Response: Thank you very much for your kind help and careful review. We have drafted a protocol of this study and it has not been published anywhere. Recently, we have submitted the protocol to register on PROSPERO. 

2. The key words on the cover page (4 of them) are more than the key words listed in the search strategy. Is there any reason that key words used in the search are only 2? If there were more key words, please consider including them in the methods. How were you able to ensure that all the relevant articles were included with using only the 2 keys words? From fig 1 flow chart, 929 TOTAL articles were identified from the 3 databases. Is it possible to provide details about the total number of articles identified in each data base after key word search and how many were excluded after the review of the titles of these studies?

Response: Thank you for your valuable advice on the revision of the flow chart. We tried to avoid omissions of key studies when we made the search strategies, thus, the key words used in the search were only 2, not including “RCT”. The studies not comparing beclomethasone-formoterol with ICS-LABA or not extrafine particles would be excluded, as was shown in the flow chart. We have made a more detailed description of the studies included and excluded, including the total number of articles identified in each data base after key word search and the number of articles excluded after the review of the titles. Some data were shown in S2 File. ( Line 173-174, 467)

3. Line 45 of the abstract results "didn't"- Please consider "did not"

Response: We have revised the words in the resubmitted manuscript.( Line 45)

4. Line 74 of the introduction: "These years, various drug formulations and inhalation techniques have been developed..." - It is not very clear what these years mean?

Response: The first ICS-LABA was developed in 2001, thus "these years" means about twenty years. We have added the detailed year information in the resubmitted manuscript. (Line 73)

5. Line 80 of the introduction "pressurized metered dose inhaler (pMDI) " -Consider using only pMDI as the abbreviation has been defined above already.

Response: We have revised the words in the resubmitted manuscript. Thank you for your careful review. (Line 80)

6. Line 83 of introduction "to generate" -Please consider another word. For the context of delivering medication to a site of action, "generate" may not be the most appropriate word to use. 

Response: We have replaced "generate" with "improve". (Line 82)

7. Line 126 of the methods "and other outcome variables of" - Please check the font size of these words. They are smaller. 

Response: We have revised it and checked the font size of the whole manuscript to ensure to meet the journal's style requirements. (Line 126)

8. Line 147 & 148 of methods "The assessment of risk of biases was performed based on the methodology reported by each enrolled study following the rules specified in the Cochrane Handbook " - Why should the risk of biases be based on the methodology of each enrolled study when only RCTs were included? It may be good to elaborate a little bit more.

Response: The risk of biases have been assessed and shown in detail in Fig 2. We included only RCTs in order to ensure the quality of evidence. 

9. Line 155 of the statistical analysis "group27 " - Please check if it is a spelling mistake.

Response: We have revised it and checked the whole manuscript to avoid a spelling mistake. Thank you for your reminder. (Line 155)

10. Line 164 of the statistical analysis "We failed to evaluate publication bias with a funnel plot and Egger’s test, due to the limited number of included studies. "- Should the inability to check for publication bias not be included as a limitation of the study?

Response: We have added it in the part of limitation in our resubmitted manuscript. (Line 329-331)

11. Line 213 of the results outcomes "bodyplethysmography" - Please check the spelling.

Response: We have revised it in the resubmitted manuscript. (Line 222)

12. Discussion section: How will the fewer number of studies and by extension relatively lower number of patients affect the results?

Response: The fewer number of included studies and patients could directly affect the statistical significance of pooled results and also indirectly affect the clinical significance due to the limited population. The number of studies and patients in our study could make a review and pooled meta-analyses but could not have an absolute conclusion on some outcomes.

13. References section: Line 346, reference 1- Please consider including the date you accessed this reference via the website.

Response: We have added the date in the resubmitted manuscript. (Line 349)

14. Appendix: Titles of figures 3,4,& 5: These titles need to be revised slightly. From the way it is written, the respective outcomes have effects on the treatment.

Response: We have revised it in the resubmitted manuscript. (Line 468-473)

Other changes: 

1. “Extrafine HFA-beclomethasone-formoterol vs nonextrafine combination of an inhaled corticosteroid and a long acting β2-agonist in patients with persistent asthma: a systematic review and meta-analysis” were corrected as “Extrafine HFA-beclomethasone-formoterol vs. nonextrafine combination of an inhaled corticosteroid and a long acting β2-agonist in patients with persistent asthma: a systematic review and meta-analysis”.(Line 1)

2. “Extrafine HFA-beclomethasone-formoterol vs. nonextrafine combination in asthma” were corrected as “Extrafine HFA-beclomethasone-formoterol vs. nonextrafine combination in asthma” .(Line 15)

3. “Among current therapies, beclometasone dipropionate-formoterol fumarate (BDP-F) fixed combination delivers extrafine particles” were corrected as “Among current therapies, BDP-F fixed combination delivers extrafine particles ”(Line 59) 

4. “The last search was performed on October 2020.” were corrected as “The last search was performed on June 2021.”(Line 113) 

5. “In Bulac 2015” were corrected as “In Bulac 2015”(Line 230) 

6. “As we know, this was the first review and meta-analysis comparing extrafine beclometasone dipropionate-formoterol fumarate (BDP-F) fixed combination via pMDI with other ICS-LABAs.” were corrected as “As we know, this was the first review and meta-analysis comparing extrafine BDP-F fixed combination via pMDI with other ICS-LABAs.”(Line 269) 

We hope that the revised version of the manuscript is now acceptable for publication in your journal. If there is anything else we should do, please do not hesitate to let us know.

I look forward to hearing from you soon.

Thank you very much.

Yours sincerely,

Chuntao Liu

---

## [Editor Report · Decision Letter 1]

22 Jul 2021

PONE-D-20-35965R1

Extrafine HFA-beclomethasone-formoterol vs. nonextrafine combination of an inhaled corticosteroid and a long acting β2-agonist in patients with persistent asthma: a systematic review and meta-analysis

PLOS ONE

Dear Dr. Liu,

Thank you for submitting your revised manuscript to PLOS ONE. After careful consideration, I am pleased with the revision provided in response to the majority of the reviewer' comments. But I have the following three minor comments in follow up on replies you provided.

- Please follow up on your addition about the results of the subgroup analysis you provided with details of the intended comparisons in the Methods part of the manuscript. Currently, I do not believe the Methods of the manuscript are including any of the subgroup analysis plans.

- Also, please elaborate on the lack of the GRADE assessment in the limitation of the manuscript, including the justification.

- Similarly, please elaborate in the Discussion part of the manuscript about the two issues of (i) what are the serious adverse drug reactions and why not identifying them, and (ii) why only including RCTs in the analysis and the potential associated risk of bias.

We look forward to receiving your revised manuscript.

Kind regards,

Daoud Al-Badriyeh

Academic Editor

PLOS ONE
---

## [Author Response · Author response to Decision Letter 1]

3 Aug 2021

Dear editor,

Thank you for your comments concerning our manuscript entitled “Extrafine HFA-beclomethasone-formoterol vs. nonextrafine combination of an inhaled corticosteroid and a long acting β2-agonist in patients with persistent asthma: a systematic review and meta-analysis” (ID: PONE-D-20-35965R1). Those comments are valuable and very helpful for revising and improving our paper. We have carefully revised the manuscript according to your comments, and the amendments are highlighted in yellow in the revised manuscript. Point by point responses to your comments are as flowing.

Responses to editor 

1. Please follow up on your addition about the results of the subgroup analysis you provided with details of the intended comparisons in the Methods part of the manuscript. Currently, I do not believe the Methods of the manuscript are including any of the subgroup analysis plans.

Response: Dear editor, thank you for your reminder. According to your suggestion, we have added the subgroup analysis plans in the part of Methods. (Line 172-173)

2. Also, please elaborate on the lack of the GRADE assessment in the limitation of the manuscript, including the justification.

Response: Thank you very much for your kind help! We have added the GRADE assessment in the Table 2. (Line 155-157, 229, 230, 273, 274, 283, 284, Page 16-17)

3. Similarly, please elaborate in the Discussion part of the manuscript about the two issues of (i) what are the serious adverse drug reactions and why not identifying them, and (ii) why only including RCTs in the analysis and the potential associated risk of bias.

Response: Thank you for your valuable advice. We think the elaborate of two issues can be placed in the method section and we add this in the Method part of manuscript. (Line 121, 122, 135-138)

Journal Requirements

Response: We have reviewed our reference list and ensured all reference were complete and correct.

Other changes: 

1. “the clinical effects of extrafine HFA-beclomethasone-formoterol (BDP-F) versus equipotent doses of nonextrafine combination of an inhaled corticosteroid and a long acting β2-agonist (ICS-LABA) in asthma.” were corrected as “the clinical effects of extrafine hydrofluoroalkane (HFA)-beclomethasone-formoterol (BDP-F) versus equipotent doses of nonextrafine combination of an inhaled corticosteroid and a long acting β2-agonist (ICS-LABA) in asthma.”.(Line 27)

2. “no significant benefit of extrafine HFA-BDP-F over nonextrafine ICS-LABA in improving Asthma Control Test score (p = 0.30)” were corrected as “no significant benefit of extrafine HFA-BDP-F over nonextrafine ICS-LABA in improving Asthma Control Test (ACT) score (p = 0.30) ” .(Line 44)

3. “the first FDCs containing extrafine formulation of BDP and formoterol (BDP-F) both in the pMDI and dry powder inhaler (DPI) versions” were corrected as “the first FDCs containing extrafine formulation of BDP and BDP-F both in the pMDI and dry powder inhaler (DPI) versions ”(Line 80) 

4. “compared to those of fluticasone-salmeterol (FP-S) DPI and budesonide-formoterol (BUD-F) DPI” were corrected as “compared to those of FP-S DPI and BUD-F DPI.”(Line 85) 

5. “a significant decrease in FeNO levels were observed in extrafine BDP-F” were corrected as “a significant decrease in fractional exhaled nitric oxide (FeNO) levels were observed in extrafine BDP-F”(Line 95, 96) 

6. “compare the clinical effects and tolerability profile of treatments with extrafine HFA-beclomethasone-formoterol (BDP-F) versus equipotent doses of nonextrafine ICS-LABA” were corrected as “compare the clinical effects and tolerability profile of treatments with extrafine HFA-BDP-F versus equipotent doses of nonextrafine ICS-LABA”(Line 105) 

7. “a significant decrease in FeNO levels were observed in extrafine BDP-F” were corrected as “a significant decrease in fractional exhaled nitric oxide (FeNO) levels were observed in extrafine BDP-F”(Line 95, 96)

8. “Airway inflammation: fractional exhaled nitric oxide (FeNO, ppb) and alveolar nitric oxide (CAlv, ppb); Asthma control: Asthma Control Test score” were corrected as “Airway inflammation: FeNO (ppb) and alveolar nitric oxide (CAlv, ppb); Asthma control: Asthma Control Test (ACT) score”(Line 130, 131)

9. “Patients were randomized to either extrafine HFA-beclomethasone/formoterol (BDP-F) via pMDI (FOSTER®; Chiesi Farmaceutici, Italy) [18-25, 30, 31] or nonextrafine ICS-LABA, including fluticasone/salmeterol (FP/S) via DPI (Seretide® Diskus®; GlaxoSmithKline, UK)[18-20, 23, 31], FP/S via pMDI (Seretide®; GlaxoSmithKline, UK) [21, 24], budesonide/formoterol (BUD/F) via DPI (Symbicort® Turbuhaler®; AstraZeneca, Sweden) [20, 22, 25] and CFC-BDP” were corrected as “Patients were randomized to either extrafine HFA-BDP-F via pMDI (FOSTER®; Chiesi Farmaceutici, Italy) [18-25, 31, 32] or nonextrafine ICS-LABA, including FP-S via DPI (Seretide® Diskus®; GlaxoSmithKline, UK)[18-20, 23, 32], FP/S via pMDI (Seretide®; GlaxoSmithKline, UK) [21, 24], BUD-F via DPI (Symbicort® Turbuhaler®; AstraZeneca, Sweden) [20, 22, 25] and Chloroflurocarbon (CFC)-BDP ”(Line 195-200)

10. “laboratory parameters and ECG” were corrected as “laboratory parameters and electrocardiogram (ECG)”(Line 288)

11. “an elevation of RV or RV/TLC ratio” were corrected as “an elevation of residual volume (RV) or RV/total lung capacity (TLC) ratio ”(Line 326, 327)

12. We have add one reference. (Line 453-455)

We hope that the revised version of the manuscript is now acceptable for publication in your journal. If there is anything else we should do, please do not hesitate to let us know.

I look forward to hearing from you soon.

Thank you very much.

Yours sincerely,

Chuntao Liu

---

## [Editor Report · Decision Letter 2]

24 Aug 2021

Extrafine HFA-beclomethasone-formoterol vs. nonextrafine combination of an inhaled corticosteroid and a long acting β2-agonist in patients with persistent asthma: a systematic review and meta-analysis

PONE-D-20-35965R2

Dear Dr. Liu,

We’re pleased to inform you that your manuscript has been judged scientifically suitable for publication and will be formally accepted for publication once it meets all outstanding technical requirements.

Kind regards,

Daoud Al-Badriyeh

Academic Editor

PLOS ONE
---

## [Editor Report · Acceptance letter]

26 Aug 2021

PONE-D-20-35965R2 

Extrafine HFA-beclomethasone-formoterol vs. nonextrafine combination of an inhaled corticosteroid and a long acting β2-agonist in patients with persistent asthma: a systematic review and meta-analysis 

Dear Dr. Liu:

I'm pleased to inform you that your manuscript has been deemed suitable for publication in PLOS ONE. Congratulations! Your manuscript is now with our production department. 

Kind regards, 

on behalf of

Dr. Daoud Al-Badriyeh 

Academic Editor

PLOS ONE